# C4 Bacterial Volatiles Improve Plant Health

**DOI:** 10.3390/pathogens10060682

**Published:** 2021-05-31

**Authors:** Bruno Henrique Silva Dias, Sung-Hee Jung, Juliana Velasco de Castro Oliveira, Choong-Min Ryu

**Affiliations:** 1Graduate Program in Genetics and Molecular Biology, Institute of Biology, University of Campinas (UNICAMP), Campinas 13083-970, SP, Brazil; bhsdias@hotmail.com; 2Brazilian BioRenewables National Laboratory (LNBR), Brazilian Center for Research in Energy and Materials (CNPEM), Campinas 13081-100, SP, Brazil; juliana.velasco@lnbr.cnpem.br; 3Molecular Phytobacteriology Laboratory, Korea Research Institute of Bioscience & Biotechnology (KRIBB), Daejeon 34141, Korea; shjung89@kribb.re.kr; 4Biosystems and Bioengineering Program, University of Science and Technology, Daejeon 34141, Korea

**Keywords:** C4 bacterial volatile compounds, plant growth-promoting rhizobacteria, induced systemic resistance, 2,3-butanediol

## Abstract

Plant growth-promoting rhizobacteria (PGPR) associated with plant roots can trigger plant growth promotion and induced systemic resistance. Several bacterial determinants including cell-wall components and secreted compounds have been identified to date. Here, we review a group of low-molecular-weight volatile compounds released by PGPR, which improve plant health, mostly by protecting plants against pathogen attack under greenhouse and field conditions. We particularly focus on C4 bacterial volatile compounds (BVCs), such as 2,3-butanediol and acetoin, which have been shown to activate the plant immune response and to promote plant growth at the molecular level as well as in large-scale field applications. We also disc/ uss the potential applications, metabolic engineering, and large-scale fermentation of C4 BVCs. The C4 bacterial volatiles act as airborne signals and therefore represent a new type of biocontrol agent. Further advances in the encapsulation procedure, together with the development of standards and guidelines, will promote the application of C4 volatiles in the field.

## 1. Introduction

Bacterial volatile compounds (BVCs) are a complex mixture of lipophilic molecules (<300 Da) derived from a wide range of biosynthetic pathways [1]. The qualitative and quantitative profiles of BVCs vary with the carbon source, the growth conditions of bacteria (oxygen availability, temperature, and pH) and the interaction of bacteria with other organisms in the environment [2]. Each bacterium emits a mixed and specific profile of BVCs, which play important roles in the bacterial life cycle and its interactions with other organisms [3]. Short-chain C4 volatiles contain four carbon atoms. Given their simple chemical structure, C4 BVCs can easily permeate membranes and cuticular wax, are water soluble and act as energy sources and precursors of complex molecules [4,5,6].

According to the mVOC (microbial volatile organic compound) database [7] (http://bioinformatics.charite.de/mvoc, accessed on 31 May 2021), approximately 40 C4 volatiles have been identified in the bacterial volatilome profiles. These volatiles are of a diversified nature and belong mainly to the chemical classes of alcohols (2,3-butanediol, 2-methylpropan-1-ol, n-butanol), ketones (2-3-butanedione, acetoin, butanone), esters (γ-butyrolactone, ethyl acetate, furan), carboxylic acids (butanedioic acid, butanoic acid, 4-aminobutanoic acid), nitrogen-containing compounds (2-dimethylamino acetonitrile, pyrazine, 2-methyl-1-nitropropane), and sulfur-rich compounds (butanethiol, 2-methylthietane, 3-methylsulfanylpropan-1-ol) (Table 1).

Although a variety of C4 volatiles, with diverse industrial applications, have been identified in bacterial volatilomes, only a few studies have been conducted to understand the role of these compounds in bacterial metabolism and interaction with other organisms. The most common C4 BVCs identified to date represent products of energy metabolism and serve as sources of carbon, nitrogen, and sulfur required for the synthesis of complex molecules such as proteins, pigments, antibiotics and polyamines, among others (Table 1). Additionally, C4 BVCs such as butyric acid (butanoic acid), 2,3-butanediol (2,3-BDO), and n-butanol become an alternative electron sink for the regeneration of NAD^+^ when aerobic respiration is restricted from pyruvate [25]. The role of γ-aminobutyric acid (GABA; 4-aminobutanoic acid), putrescine (1,4-diaminobutane), and γ-butyrolactone (GBL; dihydrofuran-2(3H)-one) in bacterial fitness remains unknown. According to recent studies, in addition to serving as a feedstock, these compounds are also related to cell differentiation processes and to the formation of endospores [26,27]. The pyrazine (1,4-diazine), pyrrole (1H-pyrrole), isobutyraldehyde (2-methyl propanal), and ethyl acetate (ethyl ethanoate) has also been used by bacteria as building block chemical to be converted to biomolecules with antimicrobial properties [28]. Among the most common C4 volatiles found in the bacterial volatilome, three are produced during butanediol fermentation: acetoin (3-hydroxybutanone), diacetyl (2-3 butanedione), and 2,3-BDO. Since the identification of the role of this bacterial pathway in the prevention of cell acidification, and the recent discovery of the potential of these compounds in promoting plant growth, these three compounds have become the main targets of research (Figure 1).

In the past few years, several BVCs have been identified, and extensive research has been conducted to understand their interactions with other organisms and to determine their industrial applications. However, few advances have been made to understand the unique effect of each C4 volatile on bacterial fitness and to determine how these metabolites act on other organisms. In this review, we focus on the role of C4 bacterial volatiles, mainly 2,3-BDO and acetoin, in bacterial fitness, physiology, and interaction with plant.

## 2. Microbial 2,3-BDO Fermentation Pathway

Several Gram-negative and -positive bacteria reportedly produce 2,3-BDO via the mixed acid fermentation pathway; however, the metabolic function of 2,3-BDO in bacteria has not yet been clearly determined [29]. In the mixed acid fermentation pathway, the monosaccharide like hexoses or pentoses is initially degraded into two molecules of pyruvate, which is metabolized into other compounds. The end products of fermentation, which are mainly determined by the bacterial genome and the carbon source, include acids such as acetic, lactic, succinic and formic acids and alcohols such as ethanol and 2,3-BDO [30,31,32]. To minimize the impact against environmental acidification, many bacteria employ the mixed acid–2,3-BDO pathway, which generates relatively neutral (less acidic) end products such as acetoin and 2,3-BDO [12,33,34]. 

The production of 2,3-BDO from pyruvate involves three conversion reactions (Figure 2). First, two pyruvate molecules produced by carbohydrate metabolism are condensed into α-acetolactate, the main intermediate of this reaction, by acetolactate synthase (ALS). Then, α-acetolactate is anaerobically decarboxylated by α-acetolactate decarboxylase (ALDC), generating R-acetoin, which is reversibly converted to 2,3-BDO through an NADH/NAD^+^ regeneration process catalyzed by acetoin reductase (AR)/2,3-butanediol dehydrogenase (BDH). Less frequently, because of instability under aerobic conditions, a spontaneous nonenzymatic reaction converts α-acetolactate into diacetyl [35,36], which is further reduced to acetoin by BDH or diacetyl reductase (DR), with the consumption of an NADH equivalent (Figure 2).

In nature, 2,3-BDO is produced as a racemic mixture of three stereoisomers, namely 2R,3R-BDO, 2R,3S-BDO, and 2S,3S-BDO. The ratio of these stereoisomers in the racemic mixture is extremely dependent on the genetic profile of the microorganism since some microorganisms possess multiple BDO conversion enzymes as well as a single bifunctional enzyme [37,38,39,40]. 

For example, *Bacillus* spp. and *Paenibacillus* spp. possess enzymes that favor the production of 2R,3R-BDO and 2R,3S-BDO isoforms, whereas members of the genera *Klebsiella* and *Enterobacter* have enzymes that favor the formation of 2S,3S-BDO and 2R,3S-BDO. On the other hand, *Pseudomonas* spp. encode enzymes that make it possible to produce all three stereoisomeric forms of 2,3-BDO [41,42,43,44,45].

Since bacterial 2,3-BDO is eco-friendly and an economical alternative to high-value-added chemicals, it serves extensive purposes in the chemical industry [46,47,48,49]. Nonetheless, little is known about the metabolic function of 2,3-BDO in microorganisms. Few studies have shown that this compound is associated with the maintenance of phytosanitary conditions, either by enhancing plant growth or by inhibiting plant pathogens.

## 3. Beneficial Effects of C4 BVCs on Plant Growth

Rhizosphere is one of the most complex ecosystems on earth, with a high heterogeneity of microenvironments, in which a gram of soil can possess more than 10^11^ bacterial cells [50,51,52]. Through an interspecific and harmonic ecological relationship, certain rhizosphere bacteria are able to improve plant health by synthesizing plant hormones, regulating ethylene production, increasing phosphorous uptake, nitrogen and iron availability and preventing the deleterious effects of pathogenic microorganisms. Recently, BVCs have been receiving increased attention, as they represent an important mode of plant growth promotion and a means of biological control against pathogens (Table 1) [53,54,55].

Due to their gaseous properties, BVCs play an important role as signaling molecules and can lead to the activation of a number of signals that regulate various physiological processes affecting plant health and growth (Figure 1) [1,56,57]. The first evidence supporting the role of BVCs as plant fitness-promoting bioactive chemicals was generated in 2003, when Ryu and coworkers showed that volatiles released by *Bacillus subtilis* strain GB03 (recently reclassified as *Bacillus amyloliquefaciens*) increased leaf surface area in the model plant *Arabidopsis thaliana*. Gas chromatography–mass spectrometry (GC-MS) analysis, together with and without headspace-solid phase microextraction (HS-SPME) revealed that 2,3-BDO is the main volatile compound responsible for the increase in leaf surface area and plant immunity in *Arabidopsis*, after validation and application of 1 and 100 μg of the pure compound [58,59]. This study, in addition to demonstrating for the first time the plant growth-promoting effect of 2,3-BDO and its dose–response relationship, opened new possibilities for the use of 2,3-BDO as a biostimulant and bioprotectant in crop production.

Organic acids released into the rhizosphere by plants and microorganisms form soluble complexes with metals, which contribute to the lowering of pH by releasing hydrogen or hydroxyl ions [60,61,62]. The acidic pH and low oxygen availability in the rhizosphere make it an ideal microenvironment for the activation of the BDO fermentation pathway by microorganisms. Since the discovery of the beneficial effect of 2,3-BDO on leaf size in *Arabidopsis*, some of the most efficient plant growth promoting rhizobacteria (PGPR) have been identified as producers of a mixture of acetoin and 2,3-BDO, which enhance plant fitness [63,64,65].

In another study, acetoin itself, also known as 3-hydroxy-2-butanone and produced by *Bacillus vallismortis* strain EXTN-1, promoted plant growth and induced systemic resistance against *Pectobacterium carotovorum* subsp. *carotovorum* strain SCC1 in tobacco (*Nicotiana tobaccum*) [66]. In tobacco, exogenous application of acetoin promoted plant growth in a concentration-dependent manner.

## 4. C4 BVCs as Biological Control Agents

In addition to promoting plant growth, C4 BVCs directly inhibit fungal growth and indirectly activate plant immunity by modulating hormone crosstalk [33,58,59,67]. Intriguingly, only methyl ethyl ketone has been shown to directly inhibit the growth of plant pathogenic fungi [33], while most of the C4 BVCs have been indirectly shown to activate plant systemic resistance following application on plant organs.

In 2006, Han et al. [67] demonstrated that 2R,3R-BDO, when applied to tobacco plants at a dose ranging from 1 mg to 100 pg per plant, enhanced aerial growth and triggered induced systemic resistance (ISR) against the soft rot pathogen, *P. carotovorum* (formerly *Erwinia carotovora*) (Table 2). Interestingly, no change in plant phenotype was observed upon the application of 2S,3S-BDO. Cho et al. [68] showed that 2R,3R-BDO released by *Pseudomonas chlororaphis* strain O6 is involved in the induction of systemic drought tolerance in *Arabidopsis*, and this trait is associated with the increased accumulation of salicylic acid (SA), which modulates stomatal closure by interfering with the abscisic acid (ABA) signaling pathway. Similar results were obtained with acetoin; exogenous application of acetoin triggered ISR against *Pseudomonas syringae* pv. *tomato* DC3000 in *Arabidopsis* [69]. Together, these studies elucidated that 2,3-BDO and its precursor acetoin trigger ISR in *Arabidopsis* through the NONEXPRESSOR OF PATHOGENESIS-RELATED (PR) GENES 1 (NPR1), a master regulator of SA signaling, and through the accumulation of SA and ethylene (ET), and the accumulation of jasmonic acid (JA) is not essential for triggering ISR (Figure 1). Subsequently, Cortes-Barco et al. [70] showed that 2R,3R-BDO triggers ISR in *Nicotiana benthamiana* against the hemibiotrophic fungus *Colletotrichum orbiculare* (Table 2). Through the analysis of plant gene expression, the authors showed that different chemical compounds trigger ISR in *N. benthamiana* through different mechanisms, and unlike the findings in *Arabidopsis*, 2R,3R-BDO activated the JA/ET-dependent signaling pathway in *N. benthamiana*. Treatment with 2R,3R-BDO also activated the JA/ET-dependent signaling pathway in the monocot *Agrostis stolonifera* against three fungal pathogens [71], although gene expression analysis revealed that 2,3-BDO induced other genes involved in the activation of the JA/ET-dependent signaling cascade, unlike what had been previously reported in other monocots including corn (*Zea mays*), rice (*Oryza sativa*) and wheat (*Triticum aestivum*) [72,73].

Meanwhile, the maize treated with 2,3-BDO had increased resistance against an airborne pathogenic fungus *Setosphaeria turcica* [74]. According to the study of Yi et al. [75], plant defense against a soil-borne pathogenic fungus *R. solani* was induced when 2,3-BDO was treated into bentgrass. The transcriptomic analysis of this study revealed that JA signaling-related genes and PR gene 5 receptor kinase are significantly upregulated by 2,3-BDO application (Table 2). Another C4 BVC 2-butanone activated systemic resistance against aphid and *Pseudomonas syringae* pv. *lachrymans* on cucumbers under open field conditions (Table 2). The JA indicator gene, *CsLOX* was increased in response to 2-butanone compared to control [76]. In the case of the plant exposed to BVCs, when the *Arabidopsis* was exposed to BVCs of *B. amyloliquefaciens* GB03 containing 2,3-BDO mostly, the elevated glucosinolates presented plant protection against a chewing insect *Spodoptera exigua* [77]. BVCs treatment also induced resistance to a necrotrophic fungal pathogen *Botrytis cinerea* in *Arabidopsis* [78]. Through the gene expression analysis by qRT-PCR, the SA responsive *PR1* and ET and JA responsive *PDF1.2* genes exhibit increased expression level by BVCs (Table 2). Under the miniature greenhouse condition, the cucumber exposed to BVCs had induced systemic resistance against *P. syringae* pv. *lachrymans* through activation of JA signaling pathway [79]. 

Another indirect effect of 2,3-BDO and acetoin on plant fitness is rhizosphere acidification, which plays a crucial role in nutrient acquisition by plants under stress conditions and microbiota modulation in the rhizosphere. Changes in soil pH in the plant rhizosphere determine the availability and therefore the absorption of micronutrients. In addition, soil pH is crucial for defining microbial interactions in the plant rhizosphere. Zhang et al. [81] showed that BVCs emitted by *B. amyloliquefaciens* strain GB03 not only significantly acidified the rhizosphere of *Arabidopsis* plants but also directly activated iron uptake by enhancing root proton release and subsequently increasing mineral uptake under iron-deficient conditions. *Bacillus* volatiles had a similar effect on iron and selenium uptake in *Arabidopsis* under alkaline conditions [81,82]. While the exogenous application of pure acetoin and 2,3-BDO upregulated cellular signaling for iron uptake through proton release, both BVCs also induced the cell-signaling of SA, nitric oxide and hydrogen peroxide, which is crucial for plant survival under abiotic stress conditions [68,83,84]. Taken together, these findings not only demonstrate how 2,3-BDO and acetoin induce plant resistance but also provide new insights into the changes in rhizosphere conditions that facilitate plant development. 

The newest insight into how 2,3-BDO contributes to plant health is the induction of root exudate secretion. Yi et al., (2016) showed that the application of 2,3-BDO elicits the release of root exudates [85]. Although the authors could not identify the compounds released from roots, it was evident that 2,3-BDO treatment, in addition to inducing ISR in pepper (*Capsicum annuum* L.), also induced the secretion of root exudates, which act to promote plant growth and modulate the rhizosphere microbiota (Figure 1). The data also demonstrated that microorganisms that carry the cluster of genes involved in 2,3-BDO and acetoin biosynthesis were able to survive in the rhizosphere longer than those that lack this gene cluster, showing that 2,3-BDO acts as a cell protectant against harmful compounds, as mentioned above.

Evidence from many studies indicates that 2,3-BDO and acetoin modulate a network of metabolic events involved in triggering hormonal responses aimed at protecting plants. However, most of these studies have been carried out under laboratory in vitro conditions, which may differ from field conditions in terms of the rhizosphere dynamics. The first report demonstrating that 2,3-BDO protects plants under field conditions was recently published. Kong et al., (2018) [80] demonstrated that treatment with 2R,3R-BDO and 2R,3S-BDO reduced the severity of disease caused by the naturally occurring plant viruses and increased the yield of mature pepper fruits (Table 2). In addition to validating the plant protective effect of 2,3-BDO in the field trials, the authors also demonstrated that different stereoisomers of 2,3-BDO exhibit distinct effects on ISR against diverse phytopathogenic viruses, suggesting that plants have distinct receptors for each of these isomers. 

Another bacterial C4 volatile that has received considerable attention for its effects on plants is GABA, a non-protein amino acid. Although the role of GABA in bacterial fitness remains unclear, studies show that this metabolite acts as a carbon and nitrogen feedstock and as a primary energy source. Additionally, GABA prevents cellular acidification and regulates nitrogen fixation in rhizobia-associated vegetable crops [86,87,88,89]. While 2,3-BDO has been widely studied to understand its impact on ISR in plants, GABA has been extensively investigated as a signaling molecule that controls stress tolerance, mediates plant growth and plant–microbe interactions and regulates pH and ion channels [90,91,92]. Exogenous GABA application has been shown to improve the performance of different plant species under stress conditions [93,94,95,96]; however, the mechanisms that regulate GABA activity remain poorly understood. 

GABA has been studied for approximately 60 years in mammalian cells as a potential inhibitor of neurotransmitters, but investigation of the role of this BVC in plants started in the early 2000s [97,98,99]. Several reviews have highlighted the importance of GABA in improving plant fitness and in regulating biotic and abiotic stress responses [88,100,101,102]. However, how GABA triggers the stress response, and how plants perceive GABA and initiate signal transduction remain unclear. Several mechanisms have been suggested in the literature concerning GABA and its effects on plants. First, GABA increases the plant photosynthetic rate, as it is a source of carbon and nitrogen, which are directly diverted to the tricarboxylic acid (TCA) cycle, providing metabolites that produce energy and accelerate plant growth [103,104]. Second, GABA upregulates osmoprotectant and antioxidant enzymes, which accelerate the metabolism of reactive oxygen species (ROS) in the chloroplast and maintain the permeability of the cell membrane during stress [105,106]. Third, GABA acts as a signaling molecule and modulates the activity of enzymes involved in nitrogen metabolism, nitrate uptake, phytohormone biosynthesis, and flavonoid biosynthesis [107,108,109,110].

Based on the accumulated data for exogenous application of bacterial volatile itself and volatile-emitting bacteria that improve plant health, it is increasingly evident that the relationship between the volatile and the response in the target organism occur in a species-specific manner and is determined by the genetic characteristics of each target organism. This establishes that C4 volatiles act differently in each organism to affect resistance and activate ISR in plants, based on the species-specific relationship between the plant and its pathogen. The studies at the molecular and functional levels are needed to better understand the mechanisms associated with bacterial volatile interactions. The biggest challenges in this area of research include the following: (i) identifying specific volatile receptors capable of triggering an immune response or promoting growth in plants; (ii) mapping the plant signaling cascade that responds to bacterial volatiles; (iii) implementing a sustainable approach for the use of BVCs in the open field to improve crop health and production; (iv) fully understanding the dynamics between volatiles and plant–soil–microorganism interactions; (v) better understanding the application of volatiles in crop protection in a cost-effective and efficient manner; (vi) determining how to produce BVCs on a large scale through an economical and a sustainable approach. In the next section, we discuss the genetic engineering and synthetic biology approaches that have been studied to tackle the last challenge.

## 5. C4 Conjugants

In addition to C4 BVCs, their chemical conjugants are also highly valuable because their effect is compatible with that of C4 BVCs. The best example is butyric acid and its conjugants [111,112]. Butyric acid, a short-chain fatty acid, is a C4 BVC produced by various species of anaerobic bacteria species such as *Clostridium* spp., *Fusobacterium nucleatum*, *Porphyromonas gingivalis*, and *Prevotella* spp. [111,112]. Examples of butyric acid conjugants include β-aminobutyric acid (BABA) and GABA [113,114]. The beneficial effects of butyric acid conjugants on plants are well-established; however, their capacity to promote plant growth or trigger ISR against plant pathogens have not yet been reported [113,115]. 

The augmentation of plant growth by indole-3-butyric acid (IBA), which a conjugant between the auxin precursor indole and butyric acid, has been studied for a long time [116,117]. Plant cuttings exogenously treated with IBA showed a significant increase in adventitious root formation compared with those treated with indole-3-acetate (IAA) in many plant species [118]. Many plant species are able to synthesize IBA [119,120,121,122]. In maize, IBA is generated from IAA in the leaves of seedlings grown in the dark [123]. In *Arabidopsis* and trees such as elm and hazelnut, IBA is also synthesized from IAA [124,125]. Interestingly, the production of ET and its precursor is increased in cuttings treated with IBA [126]. However, increase in the level of ET does not correlate with plant growth since ET suppresses plant growth. The relation between IBA-mediated growth promotion and ET level must be investigated further in the near future. IBA is produced not only plants but also by soil bacteria such as *Azospirillum brasilense*, which secretes IBA into the culture medium [127]. The supernatant containing IBA secreted by *A. brasilense* promotes lateral root formation in maize seedlings.

Another butyric acid conjugant is the non-protein amino acid BABA, an isomer of GABA. The protective effect of BABA against pathogens has been reported in many plant species including pea (*Pisum sativum*), tobacco, cotton (*Gossypium hirsutum*), pepper, cucumber (*Cucumis sativus*), and tomato (*Solanum lycopersicum*) [128,129,130,131,132,133]. However, the synthesis of BABA in plants was proved only recently [134]. Previously, BABA was considered to be generated by plant-associated microbes such as bacteria, fungi, viruses, nematodes and oomycetes [128,130,131,135,136]. Since BABA shows no direct antagonistic effects on such microbes in the culture media, the BABA-mediated plant protection can be applied to the leaves by spraying [114,128,137,138]. Meanwhile, the other study suggested BABA triggers ISR in plants through an SA-independent pathway [139]. 

Additionally, several studies have also demonstrated the acquisition of tolerance against abiotic stresses such as salinity, drought and heat in plants treated with BABA (Figure 1) [140,141,142]. Jakab et al., (2005) showed that under salinity and drought stress, the loss of water from *Arabidopsis* seedlings pretreated with BABA was significantly lower compared with the control, thus delaying the onset of wilting [140]. In addition to drought stress tolerance, heat stress tolerance was also induced in *Arabidopsis* seedlings grown in media containing 0.5 mM BABA [142].

Previously, BABA was reported to exist in the natural form only in root exudates of plants grown in soil [143]. However, the possibility of microbial contamination of root exudates could not be excluded because common plant-associated microbes also secrete BABA. Subsequently, discovery of the *Impaired in BABA-induced Immunity 1* (*IBI1*) mutant led to the identification of the BABA receptor in *Arabidopsis* [144]. Recently, it was shown that BABA is naturally present in plants under stress conditions [134]. Thevenet et al., (2017) reported endogenous BABA in plants exposed to biotrophic, necrotrophic and hemibiotrophic pathogens, salinity, and flooding [134]. 

Studies on the mechanism and application of C4 conjugants on a large scale are still in their infancy, despite the continued investigation of C4 conjugant-elicited enhancement of plant growth and ISR. Therefore, these compounds and their application in the field need to be evaluated in the future. 

## 6. Production Promotion of C4 BVCs through Metabolic Engineering

Due to the worldwide concern for replacing fossil fuels with renewable and eco-friendly products, BDO has become an interesting substrate for obtaining bioproducts with diverse applications in chemical, cosmetic, agricultural, pharmaceutical and food industries [145]. Most substances that 2,3-BDO can be converted into are currently obtained from petroleum. However, the use of petroleum as a raw material is threatened by the permanent exhaustion of this nonrenewable resource and even by political problems.

The three stereoisomers of 2,3-BDO occur as colorless and odorless liquids or crystals. Despite their structural similarity, the 2,3-BDO stereoisomers exhibit distinct physicochemical properties, which makes their separation processes difficult and expensive [146]. All three stereoisomers have high boiling points, ranging from 177 to 182 °C; therefore, their recovery by distillation requires the evaporation of a large volume of water [147,148]. Additionally, 2S,3S-BDO and 2R,3R-BDO exhibit low freezing points, thus allowing their use as antifreeze agents [149,150].

Industrial interests in microbial butanediol began during World War II with the shortage of raw material for the production of synthetic rubber. Due to the oil crisis, United States and Canada developed a pilot plant for converting 2,3-BDO into 1,3-butadiene using *Klebsiella oxytoca* and *Paenibacillus polymyxa* as microbial producers [12]. Since then, several reviews have been published highlighting the potential industrial applications of direct fermentation compounds as well as their polymers and derivatives [12,13,145,146,151]. 

In short, through acid-catalyzed dehydration reactions, it is possible to obtain methyl ethyl ketone (MEK), which is used as an industrial solvent for resins and lacquers as well as a fuel additive (Figure 2). As it has a higher heat of combustion than ethanol, MEK is used as an efficient fuel additive and for the production of high quality aviation fuels [152]. Catalytic dehydrogenation of 2,3-BDO produces diacetyl, a high-value-added food additive used as a flavoring agent in the food industry (Figure 2) [87,153]. Diacetyl is also a bacteriostatic agent, which inhibits the growth of a human pathogen *Mycobacterium tuberculosis* and other pathogenic microorganisms more efficiently than benzoic acid and has great potential in the cosmetics industry [47,152,154]. There is also tremendous interest in the use of 2,3-BDO for the production of polymers, printing inks, perfumes, drug carriers, explosives, and plasticizers [48,155,156,157,158]. 

Despite its diverse industrial applications, the large-scale production of 2,3-BDO is limited by some bottlenecks, mainly the optimization of yield and product recovery. To increase the yield of bacterial 2,3-BDO produced by fermentation, several genetic and metabolic engineering approaches have been tested in both its natural producers, such as members of the Enterobacteriaceae family and bacilli including Bacillus and Paenibacillus genera, and non-natural producers, such as Escherichia coli, Lactobacillus lactis, and Saccharomyces cerevisiae [145,159,160]. Genetic engineering approaches are mainly related to the overexpression of genes involved in the synthesis of BDO/acetoin and the knockout of genes encoding the acid product enzymes, which compete for the available pyruvate molecule (Figure 2) [13,161,162,163].

Guo et al., (2014) [164] demonstrated that a double mutant of *Klebsiella pneumoniae*, deleted for the genes encoding lactate dehydrogenase (LDH) and acetaldehyde dehydrogenase (ADH), exhibited accelerated fermentation, and higher 2,3-BDO yield (>100 g/L) from glucose in fed-batch culture. Knockout mutation of the *ldh* gene in *K. oxytoca* also resulted in higher 2,3-BDO yield because of the carbon flux targeting the fermentation of BDO by eliminating ethanol (byproduct), while simultaneously reducing the accumulation of acetoin (approximately 130 g/L) (Figure 2) [165]. Lactic acid is a major byproduct of fermentation in *Enterobacter* species, and the diversion of carbon flux toward BDO fermentation has been attempted to improve its yield [145]. In *Enterobacter aerogenes*, Jung et al., (2012) obtained a yield of 118.05 g/L in 54 h during fed-batch fermentation by deleting the *ldh* gene and optimizing medium composition and aeration conditions [166].

In another study, Jung et al., (2013) [167] disrupted a gene encoding a sucrose regulator (ScrR) in an *E. aerogenes* strain and improved the yield of 2,3-BDO (98.69 g/L) from sugarcane molasses in 36 h during fed-batch fermentation. In *Enterobacter cloacae*, Li et al., (2015) [163] showed that the knockout of genes that regulate the conversion of pyruvate to succinate, lactate and alcohol (*frdA*, *ldh*, and *adh*), together with the overexpression of the gene encoding galactose permease (*galP*), was able to improve the efficiency of 2R,3R-BDO production using glucose and xylose as substrates simultaneously, achieving 152.0 g/L of 2R,3R-BDO within 44 h of fed-batch fermentation (Figure 2).

Another approach commonly used to increase the yield of 2,3-BDO is related to the metabolic engineering of the BDH and glycerol dehydrogenase (GDH) enzymes (Figure 2). These NAD^+^/NADP^+^-dependent enzymes are responsible for converting the acetoin precursor into BDO, and their specificity is key in determining the purity of the racemic mixture [15,149,168]. Given their nontoxicity for human and animal and high efficiency in obtaining pure isomers, *Bacillus* spp. and *Paenibacillus* spp. exhibit a high potential for the large-scale production of BDO and represent excellent gene sources for the metabolic engineering of other organisms [145]. These bacterial species naturally produce equal amounts of 2R,3R-BDO (catalyzed by GDH) and 2R,3S-BDO (catalyzed by BDH) isomers. Simple changes in the GDH- and BDH-catalyzed pathways enable the production of pure isomers of 2,3-BDO since the high specificity of these enzymes facilitates product recovery. A mutant strain of *Bacillus licheniformis* lacking the *bdh* gene produced a high amount of D-2,3-BDO, with high optical purity [23]. By deleting the *gdh* and *acoR* genes (involved in acetoin degradation) in *B. licheniformis*, Qiu et al., (2016) [169] produced the highest titer of 2R,3S-BDO (98.0 g/L) ever reported. Similarly high levels of pure isomers were later produced by Ge et al., (2016) [151] by employing the same metabolic engineering strategy in large-scale fed-batch cultivation, thus demonstrating that high product concentration, with high optical purity and productivity, can be obtained in *B. licheniformis* through the modification of stereospecific enzymes BDH and GDH. In addition to these alternatives, other metabolic engineering strategies have been used in *B. subtilis* for increasing the production of BDO, for example, by manipulating NADH levels and controlling oxygen levels to redistribute the carbon flux [170,171,172]. This strategy can be adopted to enhance the production of BDO in other bacterial species.

## 7. Effect of C4 BVC on Bacterial Fitness: Case Studies with 2,3-BDO

A large number of microbial species produce BDO, but only a few species, such as those belonging to the genera *Enterobacter*, *Bacillus*, *Serratia*, *Pseudomonas* and *Klebsiella*, have been reported to produce significant amounts of this compound naturally [173,174]. Despite advances in the use of microbial 2,3-BDO for industrial purposes and plant health, it is not yet known exactly what role this compound plays in bacterial fitness. Studies conducted to date suggest that the metabolic function of BDO fermentation is closely associated with the prevention of acidification, maintenance of intracellular homeostasis and mediation of microbial interactions.

Bacterial metabolism is highly dependent on the pH, and any changes in pH directly affect the composition of the microbial population by affecting the growth of microbes through enzymatic malfunction and the production of secondary metabolites and signaling molecules [175,176,177]. Since other byproducts of anaerobic fermentation have acidic properties, the production of BDO and acetoin is believed to occur as a protective mechanism to provide an alkaline environment against oxygen-limiting or unfavorable acidic conditions [178,179]. 

Johansen and colleagues [180] were the first to show the importance of BDO fermentation in preventing extracellular acidification and improving bacterial fitness. The wild-type strain of *E. aerogenes* had better growth rate, higher activity of the 2,3-BDO pathway, and higher acetoin/BDO production at low pH than mutants defective in the production of acetoin and BDO. Due to the inability of the mutants to convert pyruvate into these neutral metabolites, it became clear that bacteria during exponential growth in an acidic environment switch their metabolic pathway to BDO fermentation to prevent acidification, a lethal condition caused by the formation of acidic compounds from pyruvate. In addition to controlling the pH, fermentation of BDO during the stationary phase, characterized by glucose exhaustion, plays a critical role in the regulation of the NAD/NADH ratio once BDO is reversibly oxidized to acetoin [180]. 

Due to the contribution of this model to the elucidation of the role of 2,3-BDO in controlling acidification stress, some studies have been conducted, and the same profile has been observed during anaerobic batch fermentation using different bacterial genera such as *Bacillus* [181], *Klebsiella* [182], *Serratia* [183], *Aeromonas* [184], and *Pseudomonas* [44]. Early studies using *Vibrio cholerae* as a model showed that the 2,3-BDO pathway is a key point during cell multiplication, as it avoids the development of an acidic environment and improves the chances of colonization in eukaryotic hosts. In 2006, Yoon and Mekalanos [185] compared three biotypes of *V. cholerae* and showed that the functional 2,3-BDO pathway is directly associated with the prevention of extracellular acidification in eukaryotic infections. The authors showed that the mutant strain capable of growing under acidic conditions could not colonize the host, corroborating the idea that under acidic conditions, bacteria switch their metabolic pathway to produce 2,3-BDO as an adaptative mechanism to neutralize the environment. Further studies have confirmed that superior pathogenic potential and growth advantage of certain strains of *V. cholerae* during the stationary phase are mainly caused by the production of a neutral fermentation end-product (i.e., 2,3-BDO) that does not inhibit bacterial growth [186,187,188]. 

In bacteria, the fate of pyruvate (whether it is converted to acetic acid or acetoin) appears to be a key determinant in the decision between life and death [189,190]. Studies on *Staphylococcus aureus* showed that BDO fermentation is strongly linked to the control of cell death and lysis because of the presence of the *alsSD* operon that encodes ALS and ALDC, which are required for acetoin production in other organisms [16,181,191,192]. Yang et al., (2006) showed that the disruption of the *alsSD* genes results in significantly reduces the stationary-phase survival rate and increases acetoin levels in *S. aureus* [192]. Later, Thomas et al., (2014) demonstrated that the *alsSD* pathway prevents *S. aureus* cell death by redirecting the intracellular carbon flux through acetoin production, which consumes protons from the cytoplasm and helps maintain the pH [190]. 

Another important role played by 2,3-BDO in bacterial fitness was reported by Hsieh and coworkers in 2007 [193]. The authors suggested that since bacteria associated with lung infection in mammals, such as *S. aureus*, *K. pneumoniae* and *Serratia marcescens*, naturally produce 2,3-BDO in significant amounts, this metabolite could be associated with host colonization. The authors reported that the direct application of 2,3-BDO in mice strains with acute lung injury had a negative regulatory effect on their innate immune response by inhibiting NF-KB signaling, which intensifies the potential of these pathogenic organisms to keep colonizing and hampering treatment. Another study demonstrated that during the inflammatory process of cystic fibrosis, bacterial 2,3-BDO directly affects the pulmonary microbiota and has a notable impact on disease progression. Venkataraman et al., (2014) showed in vitro that 2,3-BDO upregulates the global transcriptional regulator, LasR, in the pathogen *P. aeruginosa*, which controls quorum sensing, resulting in higher phenazine and exotoxin concentrations [194]. These compounds, in addition to increasing the virulence of *P. aeruginosa*, stimulate the occurrence of microorganism persistence because of their antimicrobial properties and also increase the resistance to antibiotics through biofilm formation, which inhibits the innate defense mechanisms of the host [44,195,196]. These results were further validated in vivo by Nguyen et al., (2016) [197]; the authors showed that 2,3-BDO increases the virulence of *P. aeruginosa* and modulates colonization by the respiratory tract microbiome, demonstrating that this volatile plays a critical role as a signaling molecule during microbial interactions and quorum sensing.

A recent study shed light on the complexity of the role of 2,3-BDO in improving bacterial fitness [84]. In this work, Yi et al., used three variants of *B. subtilis* strain 168 in in situ experiments: wild type (2,3-BDO natural producer), a 2,3-BDO overexpressing mutant (*pta*), and a 2,3-BDO null mutant (*pta/als*). The authors inoculated pepper roots with the three variants, and evaluated their population throughout the plant, characterized by the secretion of acidic products in the rhizosphere. At 14 days post-inoculation (dpi), the population densities of the overexpression mutant and the wild type were at least 2.8-fold higher than that of the null mutant. At 21 dpi, the null mutant could not be detected in pepper roots, and at 28 dpi, only the overexpression mutant could be detected. Interestingly, under ideal in vitro growth conditions, the mutation of the genes for 2,3-BDO overproduction and nonproduction did not affect bacterial robustness since the three variants showed similar growth patterns in vitro. Moreover, analysis of the growth kinetics of the null and overexpression mutants under different pH conditions in vitro revealed that while the null mutant was able to grow at pH 7, the overexpression mutant could grow at pH 5. Besides clearly demonstrating that BDO production has a critical effect on bacterial fitness by improving bacterial growth under acidic conditions both in vivo and in situ, Yi et al., also demonstrated that 2,3-BDO elicited peppers roots to produce unknown compounds that targeted microorganisms in a species-specific manner. While bacterial species including *E. coli*, *Ralstonia solanacearum* (wilt pathogen) and the null mutant of *B. subtilis* strain 168 and the fungus *Trichoderma* sp. were more sensitive to the exudates of pepper roots treated with synthetic 2,3-BDO, the overexpression mutant of *B. subtilis* strain 168 and the nonpathogenic biological control agent *Pseudomonas protegens* were more resistant to these root exudates compared with the wild-type *B. subtilis* strain 168. Since the genome of *P. protegens* also contains the operon for the synthesis of BDO, this result clearly indicates that 2,3-BDO nonproducers can be less fit for survival than 2,3-BDO producers in the presence of harmful root exudates. 

Although our understanding of the role of BDO fermentation in preventing intracellular acidification and in the consequent loss of cell viability and host colonization has improved over the years, more studies are needed to understand the mechanisms of 2,3-BDO production that improve bacterial fitness. Recently, studies have been mainly focused on how 2,3-BDO acts during plant–microbe interactions, further investigation is needed to fully understand why bacteria produce this compound.

## 8. Conclusions

BVCs exhibit great potential for improving plant health and fitness, as shown by 2,3-BDO and acetoin produced by the plant-associated *Bacillus* spp. In this review, we summarized additional C4 BVCs and their effects on model plant *Arabidopsis* and on agriculturally important crop plants. To maximize the biocontrol activity of C4 BVCs against plant pathogens and to improve plant growth and yield through the use of these compounds, it is important to: (1) broaden the applications of the diverse C4 BVCs; (2) understand BVC production conditions and mode of action during the activation of plant immunity; (3) conduct large-scale field trials, from seeding to harvest; (4) develop plant inoculation protocols. Chemical modification of C4 BVCs by conjugation with other bioactive compounds will be another silver bullet for overcoming the pitfalls of C4 BVC application in agriculture. Finally, the ability of C4 BVCs to modulate rhizosphere microbiota presents another opportunity for the use of bacterial volatiles on crop plants in the field. 

## Figures and Tables

**Figure 1 pathogens-10-00682-f001:**
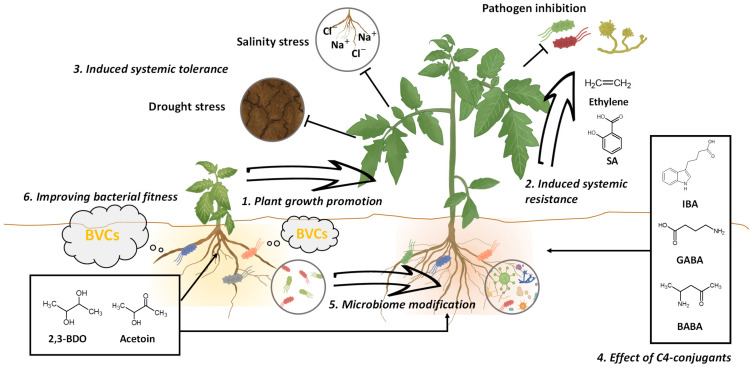
Enhancement of plant growth and stress by C4 bacterial volatile and its conjugants. 1. Plant growth promotion by C4 BVCs. BVCs such as 2,3-BDO and acetoin affect to plant health and growth in many species of plant. 2. Induced systemic resistance by BVCs. C4 BVCs trigger induced systemic resistance against plant pathogenic microbes in plants through the salicylic acid and ethylene signaling pathway. 3. Induced systemic tolerance by C4 BVCs. The BVCs exposed to plants elicits tolerance of abiotic stresses like drought, heat, and salinity stress. 4. Effect of C4 BVC-conjugants. Similar to C4 BVCs, the conjugant compounds with C4 BVCs IBA, GABA, BABA, and butyric acid have an effect on the plant growth promotion and increase of plant defense and tolerance to biotic and abiotic stresses. 5. Microbiome modification by C4 BVCs. The BVCs modulates the soil microbiome in the rhizosphere. 6. Improving bacterial fitness. C4 BVC increase own bacterial fitness under interaction with other organism like plants and animals and environment such as acidic condition.

**Figure 2 pathogens-10-00682-f002:**
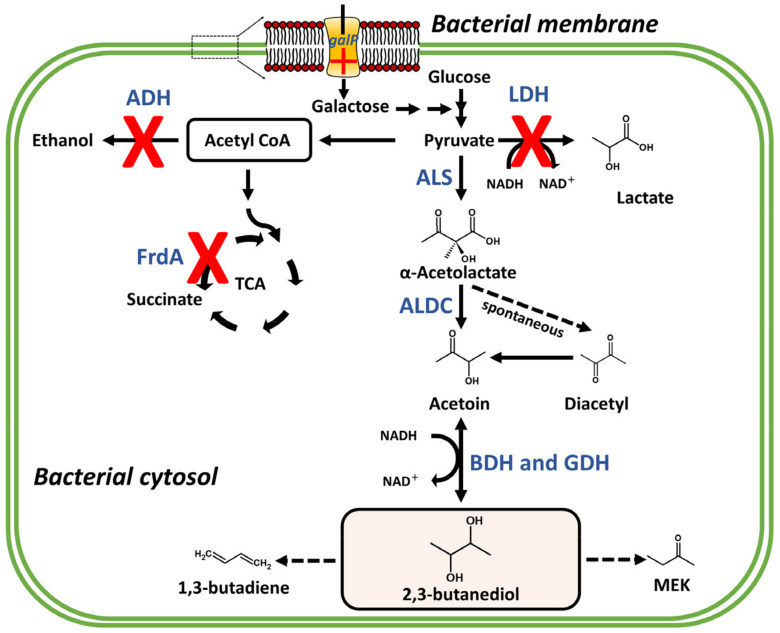
Part of butanoate metabolic pathway and genetic engineering for the high production of 2,3-butanediol and other derivatives including 1,3-butadiene and methyl ethyl ketone (MEK). Acetolactate synthase (ALS); lactate dehydrogenase (LDH); acetaldehyde dehydrogenase (ADH); α-acetolactate decarboxylase (ALDC); butanediol dehydrogenase (BDH); glycerol dehydrogenase (GDH); galactose permease (*galP*); fumarate reductase flavoprotein subunit (FrdA); the red X means mutation or deleted gene and red cross means overexpression. Blue bold letters indicate metabolic enzymes. The bold arrows indicate typical pathways and the dotted arrows represent modified pathways for industrial use.

**Table 1 pathogens-10-00682-t001:** C4 volatiles and their metabolic role in bacterial fitness.

Class	Compound	Chemical Formula	Bacterial Species	Biological Function	Reference
Acid	Succinic acid	C_4_H_6_O_4_	*Actinobacillus* spp., *Anaerobiospirillum* spp., *Bacteroides* spp., *Veillonella* spp.	Carbon feedstock, reduction of heat resistant spores, prevention of spore formation	[8]
	Butyric acid	C_4_H_8_O_2_	*Clostridium* spp., *Enterobacter* spp., *Lactobacillus* spp., *Staphylococcus* spp.	Cytokine production, carbon feedstock, electron donor	[9]
	γ-Aminobutyric acid (GABA)	C_4_H_9_NO_2_	*Xanthomonas* spp., *Streptomyces* spp.	Carbon/nitrogen feedstock, pH neutralizer	[10]
Alcohol	Isobutanol	C_4_H_10_O	*Bacillus* spp., *Enterobacter* spp., *Serratia* spp., *Streptomyces* spp., *Leuconostoc* spp.	Carbon feedstock	[11]
	Methionol	C_4_H_10_OS	*Leuconostoc* spp., *Lactobacillus* spp., *Pediococcus* spp., *Citrobacter* spp., *Alcaligenes* spp.	Carbon/sulfur feedstock	[12]
	2-3 Butanediol	C_4_H_10_O_2_	*Bacillus* spp., *Serratia* spp., *Klebsiella* spp., *Enterobacter* spp., *Pseudomonas* spp.	Electron donor, carbon feedstock, pH neutralizer, host colonization	[13]
	Butanol	C₄H₉OH	*Bacillus* spp., *Streptomyces* spp., *Paenibacillus* spp., *Ralstonia* spp., *Enterobacter* spp.	Electron donor, carbon feedstock	[14]
Ketone	Diacetyl	C_4_H_6_O_2_	*Staphylococcus* spp., *Bacillus* spp., *Klebsiella* spp., *Lactobacillus* spp., *Paenibacillus* spp.,	Electron donor, carbon feedstock, pH neutralizer	[15]
	Acetoin	C_4_H_8_O_2_	*Bacillus* spp., *Enterobacter* spp., *Pseudomonas* spp., *Serratia* spp., *Staphylococcus* spp.,	Electron donor, carbon feedstock, pH neutralizer	[16]
	Methyl ethyl ketone (MEK)	C_4_H_8_O	*Bacillus* spp., *Enterobacter* spp., *Pseudomonas* spp., *Serratia* spp., *Staphylococcus* spp.	Antifungal property	[17]
	γ-Butyrolactone (GBL)	C_4_H_6_O_2_	*Bifidobacterium* spp., *Lactobacillus* spp.	Extracellular polysaccharide production and morphological differentiation	[18]
Nitrogen-containing compound	Putrescine	C_4_H_12_N_2_	*Proteus* spp., *Yersinia* spp., *Shewanella* spp., *Ralstonia* spp.	Carbon feedstock, cell proliferation	[19]
	Pyrazine	C_4_H_4_N_2_	*Citrobacter* spp., *Klebsiella* spp., *Pseudomonas* spp.	Carbon/nitrogen feedstock, antimicrobial activities	[20]
	Pyrrole	C_4_H_5_N	*Collimonas* spp., *Serratia* spp.	Nitrogen feedstock, antibacterial effect	[21]
Aldehyde	Isobutyraldehyde	C_4_H_8_O	*Collimonas* spp., *Lactobacillus* spp., *Staphylococcus* spp.	Carbon feedstock	[22]
Ester	Ethyl acetate	C_4_H_8_O_2_	*Serratia* spp., *Collimonas* spp.	Carbon feedstock, antibacterial effect	[23]
Thioether	3-Methylthio propionate	C_4_H_8_O_2_S	*Lactococcus* spp., *Streptomyces* spp., *Chryseobacterium* spp., *Brevibacterium* spp.	Sulfur feedstock	[24]

**Table 2 pathogens-10-00682-t002:** Elicitation of induced systemic resistance by C4-BVC producing bacteria and its chemical compound.

Chemical or Emitter Bacteria	Plant Species Tested	Target Pathogens	Signaling Pathway	References
*B. amyloliquefaciens* GB03 BVCs	*Arabidopsis thaliana*	*Pectobacterium carotovorum* subsp. *carotovorum*	ET	[58]
		*Botrytis cinerea*	SA(?), ET and JA	[78]
		*Spodoptera exigua*	JA	[77]
	*Cucumis sativus* (cucumber)	*Pseudomonas syringae* pv. *lachrymans*	JA	[79]
2,3-butanediol	*Zea maize* (maize)	*Setosphaeria turcica*	NT ^1^	[74]
	*Agrostis stolonifera* (bentgrass)	*Rhizoctonia solani*	JA	[75]
2R,3R-butanediol	*Nicotiana tabaccum*	*E. carotovora* subsp. *carotovora*	NT	[67]
	*Agrostis solonifera*	*Microdochium nivale*, *R. solani* and *Sclerotinia homoeocarpa*	JA and ET	[71]
	*N. benthamiana*	*Colletotrichum orbiculare*	ET(?)	[70]
2S,3S-butanediol	*Capsicum annuum* (pepper)	CMV, TMV, PepMoV, TSWV and TYLCV ^2^	SA and JA	[80]
Acetoin	*A. thaliana*	*Pseudomonas syringae* pv. *tomato*	ET and SA	[69]
2-butanone	*C. sativus*	*Myzus persicae* (Aphid) and *P. syringae* pv. *lachrymans*	JA	[76]

^1^ NT: nontested. ^2^ CMV: cucumber mosaic virus, TMV: tobacco mosaic virus, PepMoV: pepper mottle virus, TSWV: tomato spotted wilt virus, and TYLCV: tomato yellow leaf curl virus.

## Data Availability

Not applicable.

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
