# Peer review of "C4 Bacterial Volatiles Improve Plant Health"

_pathogens, 2021, doi:10.3390/pathogens10060682_

Round 1
Reviewer 1 Report
The authors of Dias et al. very extensively characterized, grouped about 40 bacterial compounds volatile compounds (BVCs) focusing hort-chain C4 volatiles contain four carbon atom, an important common feature of which is the ease of penetration through the membranes. C4 volatiles include at least 5 chemical classes: alcohols, ketones, carboxylic acids, nitrogen-containing compounds, sulfur-rich compounds. The manuscript highlight a variety of metabolic, physiological and effects on basic processes: for aerobic respiration, cell differentiation processes and to the formation of spores.
The paper focus on three compounds produced during butanediol fermentation: acetoin (3-hy- 61 droxybutanone), diacetyl (2-3 butanedione), and 2,3-butanediol (2,3-BDO), which have the potential to interact in promoting plant growth. To minimize environmental acidification, many bacteria employ the mixed acid–2,3-BDO pathway, which generates relatively neutral (less acidic) and products such as acetoin and 2,3-BDO
Because of their gaseous properties, BVCs play an important role as signaling molecules and can lead to the activation of a number of signals that regulate various physiological processes affecting plant health and growth demonstrating for the first time the plant growth-promoting effect of 2,3-BDO and its dose–response relationship, opened new possibilities for the use of 2,3-BDO as a biostimulant and bioprotectant in crop production.
The paper give numerous examples of plant immunization against pathogens of different types by C4 volatiles synthesized by various bacteria, induction of root exudate secretion and hormonal responses, and a link with non-protein amino acid GABA in regulating osmoprotetion and resistance to other numerous abiotic stresses.
The authors repeatedly point out that C4 BVCs have a positive effect on plants by inducing immunity. However, the description of the pathways of immunity is incomplete. The authors completely overlook the very important pathway of jasmonic acid. Besides, they do not mention that the basic signalling substances in the induced resistance of plants to jasmonic acid and salicylic acid occur in the form of volatile methylated jasmonanates and methyl salicinates, in which form they can transfer between plants. There is also no clear reference to pathogen-dependent proteins arising in immunity.
No clear distinction between direct and indirect (through induction of immunity pathways) impact of C4 BVCs in operation
The text should be represented in more tables, diagrams by means of bullets.
There is too much text without bullets, figures etc
There is not enough recent literature from 2019, 2020 and 2021
Author Response
Reviewer 1: The authors repeatedly point out that C4 BVCs have a positive effect on plants by inducing immunity. However, the description of the pathways of immunity is incomplete. The authors completely overlook the very important pathway of jasmonic acid. Besides, they do not mention that the basic signaling substances in the induced resistance of plants to jasmonic acid and salicylic acid occur in the form of volatile methylated jasmonanates and methyl salicinates, in which form they can transfer between plants. There is also no clear reference to pathogen-dependent proteins arising in immunity.
Authors' Response: We appreciate the reviewer’s comments. We partially agree his/her opinion. In the text, we improved the context regarding the signaling pathways including jasmonic acid, salicylic acid, and their methylated form during elicitation of ISR in pages 6-8 and added a new Table 2. The new references were newly input. The Table 2 clearly demonstrates the pathogen- and C4 BVC-dependent immune response.
Reviewer 1: No clear distinction between direct and indirect (through induction of immunity pathways) impact of C4 BVCs in operation.
Authors' Response: We clarified the role of C4 BVC on plant protection against pathogen attack on page 6 as below:
“In addition to promoting plant growth, C4 BVCs directly inhibit fungal growth and indirectly activate plant immunity by modulating hormone crosstalk [21,58,59,67]. Intriguingly, only methyl ethyl ketone has been shown to directly inhibit the growth of plant pathogenic fungi [21], while most of the C4 BVCs have been indirectly shown to activate plant systemic resistance following application on plant organs.”
Reviewer 1: The text should be represented in more tables, diagrams by means of bullets.
Authors' Response: We thanks the comments. Following your comment, we created one Figure and one Table newly as well as we also modified the old Figure 1.
Reviewer 1: There is too much text without bullets, figures etc.
Authors' Response: As we already described, we modified manuscript succinctly and include new descriptions for new Table and Figure.
Reviewer 1: There is not enough recent literature from 2019, 2020 and 2021.
Authors' Response: We are sorry for missing recent references. Now we have 14 and 2 references that were published at 2019 and 2020 respectively. We did not find any new references that related to our topic.
Reviewer 2 Report
I write in reference to the manuscript “C4 Bacterial Volatiles Improve Plant Health” by Silva Dias. The paper is a review about the effect of the bacterial volatile compounds on plant growth and immunity and some topics of metabolic engineering, and large-scale fermentation.
In my opinion, this is a complete and well written piece of work which would surely be useful for readers interested in the topic.
I have a single suggestion. Please include a figure with a metabolic map, similar to the previously presented in the figure 3 of Ryu et al. 2003 (10.1073_pnas.0730845100) but more extensive to join the pathways, compounds (1,3-butadiene, methyl ethyl ketone, diacetyl, acetaldehyde, ethanol, Lactic acid, acetic acid) and alternative enzymes referred in the sections “6. Production of C4 BVCs through Metabolic Engineering” and “7. Effect of C4 BVC Production on Bacteria: Case Studies with 2,3-BDO."
In my opinion the work would benefit from this map because it would facilitate the following of sections 6 and 7.
The proposed figure could be partially redundant with a part of figure 1, in that case, figure 1 could be modified to avoid this problem.
Author Response
Reviewer 2: I have a single suggestion. Please include a figure with a metabolic map, similar to the previously presented in the figure 3 of Ryu et al. 2003 (10.1073_pnas.0730845100) but more extensive to join the pathways, compounds (1, 3-butadiene, methyl ethyl ketone, diacetyl, acetaldehyde, ethanol, Lactic acid, acetic acid) and alternative enzymes referred in the sections “6. Production of C4 BVCs through Metabolic Engineering” and “7. Effect of C4 BVC Production on Bacteria: Case Studies with 2,3-BDO.”
In my opinion the work would benefit from this map because it would facilitate the following of sections 6 and 7.
The proposed figure could be partially redundant with a part of figure 1, in that case, figure 1 could be modified to avoid this problem.
Authors' Response: We really appreciate the reviewer’s valuable comments. As the reviewer suggested kindly, we removed the redundant butanoate pathway in Figure 1 and created a new Figure 2 that describe the pathways and final products. The new Figure also included the information for “6. Production of C4 BVCs through Metabolic Engineering”. We also modified the Figure 1 that included comment for “7. Effect of C4 BVC Production on Bacteria: Case Studies with 2,3-BDO.”. Now the manuscript was improved and gives more information for general audiences.
Round 2
Reviewer 1 Report
The authors made a very intensive correction of the publication submitted for review. They collected in the form of a table information on the elicitor function of C4 volatile compounds, additionally marking the signal substances important in the plant resistance pathways: jasmonic acid, ethylene, salicylic acid. The most recent literature, mainly concern induced resistance, has also been completed.